# Scalable Semi-Supervised Aggregation of Classifiers

**Akshay Balsubramani**
UC San Diego
abalsubr@cs.ucsd.edu

**Yoav Freund**
UC San Diego
yfreund@cs.ucsd.edu

## Abstract

We present and empirically evaluate an efficient algorithm that learns to aggregate the predictions of an ensemble of binary classifiers. The algorithm uses the structure of the ensemble predictions on unlabeled data to yield significant performance improvements. It does this without making assumptions on the structure or origin of the ensemble, without parameters, and as scalably as linear learning. We empirically demonstrate these performance gains with random forests.

## 1 Introduction

Ensemble-based learning is a very successful approach to learning classifiers, including well-known methods like boosting [1], bagging [2], and random forests [3]. The power of these methods has been clearly demonstrated in open large-scale learning competitions such as the Netflix Prize [4] and the ImageNet Challenge [5]. In general, these methods train a large number of "base" classifiers and then combine them using a (possibly weighted) majority vote. By aggregating over classifiers, ensemble methods reduce the variance of the predictions, and sometimes also reduce the bias [6].

The ensemble methods above rely solely on a *labeled* training set of data. In this paper we propose an ensemble method that uses a large *unlabeled* data set in addition to the labeled set. Our work is therefore at the intersection of semi-supervised learning [7, 8] and ensemble learning.

This paper is based on recent theoretical results of the authors [9]. Our main contributions here are to extend and apply those results with a new algorithm in the context of random forests [3] and to perform experiments in which we show that, when the number of labeled examples is small, our algorithm's performance is at least that of random forests, and often significantly better.

For the sake of completeness, we provide an intuitive introduction to the analysis given in [9]. How can unlabeled data help in the context of ensemble learning? Consider a simple example with six equiprobable data points. The ensemble consists of six classifiers, partitioned into three "A" rules and three "B" rules. Suppose that the "A" rules each have error $1/3$ and the "B" rules each have error $1/6$. [1] If given only this information, we might take the majority vote over the six rules, possibly giving lower weights to the "A" rules because they have higher errors.

Suppose, however, that we are given the unlabeled information in Table 1. The columns of this table correspond to the six classifiers and the rows to the six unlabeled examples. Each entry corresponds to the prediction of the given classifier on the given example. As we see, the main difference between the "A" rules and the "B" rules is that any two "A" rules disagree with probability $1/3$, whereas the "B" rules always agree. For this example, it can be seen (e.g. proved by contradiction) that the only possible true labeling of the unlabeled data that is consistent with Table 1 *and* with the errors of the classifiers is that all the examples are labeled '+'.

Consequently, we conclude that the majority vote over the "A" rules has zero error, performing significantly better than any of the base rules. In contrast, giving the "B" rules equal weight would

result in an a rule with error $1/6$. Crucially, our reasoning to this point has solely used the structure of the *unlabeled examples* along with the error rates in Table 1 to constrain our search for the true labeling.

|        | A classifiers | | | B classifiers | | |
|--------|-----|-----|-----|-----|-----|-----|
| $x_1$  | -   | +   | +   | +   | +   | +   |
| $x_2$  | -   | +   | +   | +   | +   | +   |
| $x_3$  | +   | -   | +   | +   | +   | +   |
| $x_4$  | +   | -   | +   | +   | +   | +   |
| $x_5$  | +   | +   | -   | +   | +   | +   |
| $x_6$  | +   | +   | -   | -   | -   | -   |
| error  | 1/3 | 1/3 | 1/3 | 1/6 | 1/6 | 1/6 |

Table 1: An example of the utility of unlabeled examples in ensemble learning

By such reasoning alone, we have correctly predicted according to a weighted majority vote. This example provides some insight into the ways in which unlabeled data can be useful:

- When combining classifiers, diversity is important. It can be better to combine less accurate rules that disagree with each other than to combine more accurate rules that tend to agree.
- The bounds on the errors of the rules can be seen as a set of constraints on the true labeling. A complementary set of constraints is provided by the unlabeled examples. These sets of constraints can be combined to improve the accuracy of the ensemble classifier.

The above setup was recently introduced and analyzed in [9]. That paper characterizes the problem as a zero-sum game between a predictor and an adversary. It then describes the minimax solution of the game, which corresponds to an efficient algorithm for transductive learning.

In this paper, we build on the worst-case framework of [9] to devise an efficient and practical semi-supervised aggregation algorithm for random forests. To achieve this, we extend the framework to handle *specialists* – classifiers which only venture to predict on a subset of the data, and abstain from predicting on the rest. Specialists can be very useful in targeting regions of the data on which to precisely suggest a prediction.

The high-level idea of our algorithm is to artificially generate new specialists from the ensemble. We incorporate these, and the targeted information they carry, into the worst-case framework of [9]. The resulting aggregated predictor inherits the advantages of the original framework:

(A) **Efficient:** Learning reduces to solving a scalable $p$-dimensional convex optimization, and test-time prediction is as efficient and parallelizable as $p$-dimensional linear prediction.
(B) **Versatile/robust:** No assumptions about the structure or origin of the predictions or labels.
(C) **No introduced parameters:** The aggregation method is completely data-dependent.
(D) **Safe:** Accuracy guaranteed to be at least that of the best classifier in the ensemble.

We develop these ideas in the rest of this paper, reviewing the core worst-case setting of [9] in Section 2, and specifying how to incorporate specialists and the resulting learning algorithm in Section 3.

Then we perform an exploratory evaluation of the framework on data in Section 4. Though the framework of [9] and our extensions can be applied to any ensemble of arbitrary origin, in this paper we focus on random forests, which have been repeatedly demonstrated to have state-of-the-art, robust classification performance in a wide variety of situations [10]. We use a random forest as a base ensemble whose predictions we aggregate. But unlike conventional random forests, we do not simply take a majority vote over tree predictions, instead using a unlabeled-data-dependent aggregation strategy dictated by the worst-case framework we employ.

## 2 Preliminaries

A few definitions are required to discuss these issues concretely, following [9]. Write $[a]_+ = \max(0, a)$ and $[n] = \{1, 2, \dots, n\}$. All vector inequalities are componentwise.

We first consider an ensemble $\mathcal{H} = \{h_1, \ldots, h_p\}$ and unlabeled data $x_1, \ldots, x_n$ on which we wish to predict. As in [9], the predictions and labels are allowed to be randomized, represented by values in $[-1, 1]$ instead of just the two values $\{-1, 1\}$. The ensemble's predictions on the unlabeled data are denoted by $\mathbf{F}$:

$$\mathbf{F} = \begin{pmatrix} h_1(x_1) & h_1(x_2) & \cdots & h_1(x_n) \\ \vdots & \vdots & \ddots & \vdots \\ h_p(x_1) & h_p(x_2) & \cdots & h_p(x_n) \end{pmatrix} \in [-1, 1]^{p \times n} \tag{1}$$

We use vector notation for the rows and columns of $\mathbf{F}$: $\mathbf{h}_i = (h_i(x_1), \cdots, h_i(x_n))^\top$ and $\mathbf{x}_j = (h_1(x_j), \cdots, h_p(x_j))^\top$. The *true labels* on the test data $T$ are represented by $\mathbf{z} = (z_1; \ldots; z_n) \in [-1, 1]^n$. The labels $\mathbf{z}$ are hidden from the predictor, but we assume the predictor has knowledge of a *correlation vector* $\mathbf{b} \in (0, 1]^p$ such that $\frac{1}{n} \sum_j h_i(x_j) z_j \geq b_i$, i.e. $\frac{1}{n} \mathbf{Fz} \geq \mathbf{b}$. These $p$ constraints on $\mathbf{z}$ exactly represent upper bounds on individual classifier error rates, which can be estimated from the training set w.h.p. when all the data are drawn i.i.d., in a standard way also used by empirical risk minimization (ERM) methods that simply predict with the minimum-error classifier [9].

## 2.1 The Transductive Binary Classification Game

The idea of [9] is to formulate the ensemble aggregation problem as a two-player zero-sum game between a predictor and an adversary. In this game, the predictor is the first player, who plays $\mathbf{g} = (g_1; g_2; \ldots; g_n)$, a randomized label $g_i \in [-1, 1]$ for each example $\{x_i\}_{i=1}^n$. The adversary then sets the labels $\mathbf{z} \in [-1, 1]^n$ under the ensemble classifier error constraints defined by $\mathbf{b}$. [2] The predictor's goal is to minimize the *worst-case expected classification error on the test data* (w.r.t. the randomized labelings $\mathbf{z}$ and $\mathbf{g}$), which is just $\frac{1}{2}\left(1 - \frac{1}{n}\mathbf{z}^\top \mathbf{g}\right)$. This is equivalently viewed as maximizing worst-case correlation $\frac{1}{n}\mathbf{z}^\top\mathbf{g}$. To summarize concretely, we study the following game:

$$V := \max_{\mathbf{g} \in [-1,1]^n} \min_{\substack{\mathbf{z} \in [-1,1]^n, \\ \frac{1}{n}\mathbf{Fz} \geq \mathbf{b}}} \frac{1}{n}\mathbf{z}^\top \mathbf{g} \tag{2}$$

The minimax theorem ([1], p.144) applies to the game (2), and there is an optimal strategy $\mathbf{g}^*$ such that $\min_{\substack{\mathbf{z} \in [-1,1]^n, \\ \frac{1}{n}\mathbf{Fz} \geq \mathbf{b}}} \frac{1}{n}\mathbf{z}^\top \mathbf{g}^* \geq V$, guaranteeing worst-case prediction error $\frac{1}{2}(1 - V)$ on the $n$ unlabeled data. This optimal strategy $\mathbf{g}^*$ is a simple function of a particular weighting over the $p$ hypotheses – a nonnegative $p$-vector.

**Definition 1** (Slack Function). *Let $\sigma \geq 0^p$ be a weight vector over $\mathcal{H}$ (not necessarily a distribution). The vector of **ensemble predictions** is $\mathbf{F}^\top \sigma = (\mathbf{x}_1^\top \sigma, \ldots, \mathbf{x}_n^\top \sigma)$, whose elements' magnitudes are the **margins**. The **prediction slack function** is*

$$\gamma(\sigma, \mathbf{b}) := \gamma(\sigma) := -\mathbf{b}^\top \sigma + \frac{1}{n} \sum_{j=1}^n \left[ \left| \mathbf{x}_j^\top \sigma \right| - 1 \right]_+ \tag{3}$$

*and this is convex in $\sigma$. The **optimal weight vector** $\sigma^*$ is any minimizer $\sigma^* \in \arg\min_{\sigma \geq 0^p} [\gamma(\sigma)]$.*

The main result of [9] uses these to describe the minimax equilibrium of the game (2).

**Theorem 2** ([9]). *The minimax value of the game (2) is $V = -\gamma(\sigma^*)$. The minimax optimal predictions are defined as follows: for all $j \in [n]$,*

$$g_j^* := g_j(\sigma^*) = \begin{cases} \mathbf{x}_j^\top \sigma^* & \left| \mathbf{x}_j^\top \sigma^* \right| < 1 \\ \mathrm{sgn}(\mathbf{x}_j^\top \sigma^*) & otherwise \end{cases}$$

## 2.2 Interpretation

Theorem 2 suggests a statistical learning algorithm for aggregating the $p$ ensemble classifiers' predictions: estimate $\mathbf{b}$ from the training (labeled) set, optimize the convex slack function $\gamma(\sigma)$ to find $\sigma^*$, and finally predict with $g_j(\sigma^*)$ on each example $j$ in the test set. The resulting predictions are guaranteed to have low error, as measured by $V$. In particular, it is easy to prove [9] that $V$ is at least $\max_i b_i$, the performance of the best classifier.

The slack function (3) merits further scrutiny. Its first term depends only on the labeled data and not the unlabeled set, while the second term $\frac{1}{n}\sum_{j=1}^{n}\left[\left|\mathbf{x}_j^\top\sigma\right|-1\right]_+$ incorporates only unlabeled information. These two terms trade off smoothly – as the problem setting becomes fully supervised and unlabeled information is absent, the first term dominates, and $\sigma^*$ tends to put all its weight on the best single classifier like ERM.

Indeed, this viewpoint suggests a (loose) interpretation of the second term as an unsupervised regularizer for the otherwise fully supervised optimization of the "average" error $\mathbf{b}^\top\sigma$. It turns out that a change in the regularization factor corresponds to different constraints on the true labels $\mathbf{z}$:

**Theorem 3** ([9]). *Let* $V_\alpha := \displaystyle\max_{\mathbf{g}\in[-1,1]^n}\min_{\substack{\mathbf{z}\in[-\alpha,\alpha]^n,\\ \frac{1}{n}\mathbf{Fz}\geq\mathbf{b}}}\frac{1}{n}\mathbf{z}^\top\mathbf{g}$ *for any* $\alpha > 0$. *Then* $V_\alpha = \min_{\sigma\geq 0^p}\left[-\mathbf{b}^\top\sigma + \frac{\alpha}{n}\sum_{j=1}^{n}\left[\left|\mathbf{x}_j^\top\sigma\right|-1\right]_+\right]$.

So the regularized optimization assumes each $z_i \in [-\alpha, \alpha]$. For $\alpha < 1$, this is equivalent to assuming the usual binary labels ($\alpha = 1$), and then *adding uniform random label noise*: flipping the label w.p. $\frac{1}{2}(1-\alpha)$ on each of the $n$ examples independently. This encourages "clipping" of the ensemble predictions $\mathbf{x}_j^\top\sigma^*$ to the $\sigma^*$-weighted *majority vote* predictions, as specified by $\mathbf{g}^*$.

## 2.3 Advantages and Disadvantages

This formulation has several significant merits that would seem to recommend its use in practical situations. It is very efficient – once $\mathbf{b}$ is estimated (a scalable task, given the labeled set), the slack function $\gamma$ is effectively an average over convex functions of i.i.d. unlabeled examples, and consequently is amenable to standard convex optimization techniques [9] like stochastic gradient descent (SGD) and variants. These only operate in $p$ dimensions, independent of $n$ (which is $\gg p$). The slack function is Lipschitz and well-behaved, resulting in stable approximate learning.

Moreover, test-time prediction is extremely efficient, because it only requires the $p$-dimensional weighting $\sigma^*$ and can be computed example-by-example on the test set using only a dot product in $\mathbb{R}^p$. The form of $\mathbf{g}^*$ and its dependence on $\sigma^*$ facilitates interpretation as well, as it resembles familiar objects: sigmoid link functions for linear classifiers.

Other advantages of this method also bear mention: it makes no assumptions on the structure of $\mathcal{H}$ or $\mathbf{F}$, is provably robust against the worst case, and adds no input parameters that need tuning. These benefits are notable because they will be inherited by our extension of the framework in this paper.

However, this algorithm's practical performance can still be mediocre on real data, which is often easier to predict than an adversarial setup would have us believe. As a result, we seek to add more information in the form of constraints on the adversary, to narrow the gap between it and reality.

## 3 Learning with Specialists

To address this issue, we examine a generalized scenario in which each classifier in the ensemble can abstain on any subset of the examples instead of predicting $\pm 1$. It is a *specialist* that predicts only over a subset of the input, and we think of its abstain/participate decision being randomized in the same way as the randomized label on each example. In this section, we extend the framework of Section 2.1 to arbitrary specialists, and discuss the semi-supervised learning algorithm that results.

In our formulation, suppose that for a classifier $i \in [p]$ and an example $x$, the classifier decides to abstain with probability $1 - v_i(x)$. But if the decision is to participate, the classifier predicts

$h_i(x) \in [-1, 1]$ as previously. Our *only* assumption on $\{v_i(x_1), \ldots, v_i(x_n)\}$ is the reasonable one that $\sum_{j=1}^{n} v_i(x_j) > 0$, so classifier $i$ is not a worthless specialist that abstains everywhere.

The constraint on classifier $i$ is now not on its correlation with $\mathbf{z}$ on the entire test set, but on the average correlation with $\mathbf{z}$ restricted to occasions on which it participates. So for some $[b_S]_i \in [0, 1]$,

$$\sum_{j=1}^{n} \left( \frac{v_i(x_j)}{\sum_{k=1}^{n} v_i(x_k)} \right) h_i(x_j) z_j \geq [b_S]_i \tag{4}$$

Define $\rho_i(x_j) := \frac{v_i(x_j)}{\sum_{k=1}^{n} v_i(x_k)}$ (a distribution over $j \in [n]$) for convenience. Now redefine our unlabeled data matrix as follows:

$$\mathbf{S} = n \begin{pmatrix} \rho_1(x_1) h_1(x_1) & \rho_1(x_2) h_1(x_2) & \cdots & \rho_1(x_n) h_1(x_n) \\ \vdots & \vdots & \ddots & \vdots \\ \rho_p(x_1) h_p(x_1) & \rho_p(x_2) h_p(x_2) & \cdots & \rho_p(x_n) h_p(x_n) \end{pmatrix} \tag{5}$$

Then the constraints (4) can be written as $\frac{1}{n} \mathbf{S} \mathbf{z} \geq \mathbf{b}_S$, analogous to the initial prediction game (2).

To summarize, our specialist ensemble aggregation game is stated as

$$V_S := \min_{\substack{\mathbf{z} \in [-1,1]^n, \\ \frac{1}{n} \mathbf{S} \mathbf{z} \geq \mathbf{b}_S}} \max_{\mathbf{g} \in [-1,1]^n} \frac{1}{n} \mathbf{z}^\top \mathbf{g} \tag{6}$$

We can immediately solve this game from Thm. 2, with $(\mathbf{S}, \mathbf{b}_S)$ simply taking the place of $(\mathbf{F}, \mathbf{b})$.

**Theorem 4** (Solution of the Specialist Aggregation Game). *The **awake ensemble prediction** w.r.t. weighting $\sigma \geq 0^p$ on example $x_i$ is $[\mathbf{S}^\top \sigma]_i = n \sum_{j=1}^{p} \rho_j(x_i) h_j(x_i) \sigma_j$ . The slack function is now*

$$\gamma_S(\sigma) := \frac{1}{n} \sum_{j=1}^{n} \left[ \left| [\mathbf{S}^\top \sigma]_j \right| - 1 \right]_+ - \mathbf{b}_S^\top \sigma \tag{7}$$

*The minimax value of this game is $V_S = \max_{\sigma \geq 0^p} [-\gamma_S(\sigma)] = -\gamma_S(\sigma_S^*)$. The minimax optimal predictions are defined as follows: for all $i \in [n]$,*

$$[g_S^*]_i \doteq g_S(\sigma_S^*) = \begin{cases} [\mathbf{S}^\top \sigma_S^*]_i & \left| [\mathbf{S}^\top \sigma_S^*]_i \right| < 1 \\ \mathrm{sgn}([\mathbf{S}^\top \sigma_S^*]_i) & otherwise \end{cases}$$

In the no-specialists case, the vector $\rho_i$ is the uniform distribution $(\frac{1}{n}, \ldots, \frac{1}{n})$ for any $i \in [p]$, and the problem reduces to the prediction game (2). As in the original prediction game, the minimax equilibrium depends on the data only through the ensemble predictions, but these are now of a different form. Each example is now weighted proportionally to $\rho_j(x_i)$. So on any given example $x_i$, only hypotheses which participate on it will be counted; and those that specialize the most narrowly, and participate on the fewest other examples, will have more influence on the eventual prediction $g_i$, ceteris paribus.

### 3.1   Creating Specialists for an Algorithm

We can now present the main ensemble aggregation method of this paper, which creates specialists from the ensemble, adding them as additional constraints (rows of $\mathbf{S}$). The algorithm, HEDGECLIPPER, is given in Fig. 1, and instantiates our specialist learning framework with a random forest [3]. As an initial exploration of the framework here, random forests are an appropriate base ensemble because they are known to exhibit state-of-the-art performance [10]. Their well-known advantages also include scalability, robustness (to corrupt data and parameter choices), and interpretability; each of these benefits is shared by our aggregation algorithm, which consequently inherits them all.

Furthermore, decision trees are a natural fit as the ensemble classifiers because they are inherently hierarchical. Intuitively (and indeed formally too [11]), they act like nearest-neighbor (NN) predictors w.r.t. a distance that is "adaptive" to the data. So each tree in a random forest represents a

somewhat different, nonparametric partition of the data space into regions in which one of the labels $\pm 1$ dominates. Each such region corresponds exactly to a leaf of the tree.

The idea of HEDGECLIPPER is simply to **consider each leaf in the forest as a specialist**, which predicts only on the data falling into it. By the NN intuition above, these specialists can be viewed as predicting on data that is near them, where the supervised training of the tree attempts to define the purest possible partitioning of the space. A pure partitioning results in many specialists with $[b_S]_i \approx 1$, each of which contributes to the awake ensemble prediction w.r.t. $\sigma^*$ over its domain, to influence it towards the correct label (inasmuch as $[b_S]_i$ is high).

Though the idea is complex in concept for a large forest with many arbitrarily overlapping leaves from different trees, it fits the worst-case specialist framework of the previous sections. So the algorithm is still essentially linear learning with convex optimization, as we have described.

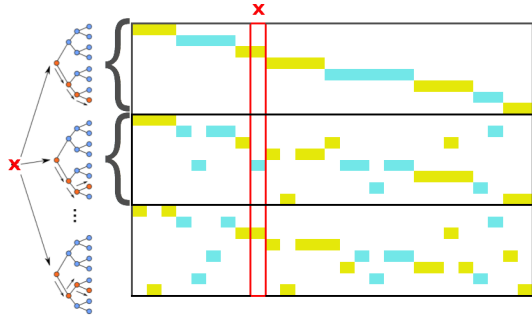

**Algorithm 1** HEDGECLIPPER

**Input:** Labeled set $L$, unlabeled set $U$
 1: Using $L$, grow trees $\mathcal{T} = \{T_1, \ldots, T_p\}$ (regularized; see Sec. 3.2)
 2: Using $L$, estimate $\mathbf{b}_S$ on $\mathcal{T}$ and its leaves
 3: Using $U$, (approximately) optimize (7) to estimate $\sigma_S^*$
**Output:** The estimated weighting $\sigma_S^*$, for use at test time

Figure 1: At left is algorithm HEDGECLIPPER. At right is a schematic of how the forest structure is related to the unlabeled data matrix $\mathbf{S}$, with a given example $x$ highlighted. The two colors in the matrix represent $\pm 1$ predictions, and white cells abstentions.

### 3.2 Discussion

Trees in random forests have thousands of leaves or more in practice. As we are advocating adding so many extra specialists to the ensemble for the optimization, it is natural to ask whether this erodes some of the advantages we have claimed earlier.

Computationally, it does not. When $\rho_j(x_i) = 0$, i.e. classifier $j$ abstains deterministically on $x_i$, then the value of $h_j(x_i)$ is irrelevant. So storing $\mathbf{S}$ in a sparse matrix format is natural in our setup, with the accompanying performance gain in computing $\mathbf{S}^\top \sigma$ while learning $\sigma^*$ and predicting with it. This turns out to be crucial to efficiency – each tree induces a partitioning of the data, so the set of rows corresponding to any tree contains $n$ nonzero entries *in total*. This is seen in Fig. 1.

Statistically, the situation is more complex. On one hand, there is no danger of overfitting in the traditional sense, regardless of how many specialists are added. Each additional specialist can only shrink the constraint set that the adversary must follow in the game (6). It only adds information about $\mathbf{z}$, and therefore the performance $V_S$ must improve, if the game is solved exactly.

However, for learning we are only concerned with *approximately* optimizing $\gamma_S(\sigma)$ and solving the game. This presents several statistical challenges. Standard optimization methods do not converge as well in high ambient dimension, even given the structure of our problem. In addition, random forests practically perform best when each tree is grown to overfit. In our case, on any sizable test set, small leaves would cause some entries of $\mathbf{S}$ to have large magnitude, $\gg 1$. This can foil an algorithm like HEDGECLIPPER by causing it to vary wildly during the optimization, particularly since those leaves' $[b_S]_i$ values are only roughly estimated.

From an optimization perspective, some of these issues can be addressed by e.g. (pseudo)-second-order methods [12], whose effect would be interesting to explore in future work. Our implementation opts for another approach – to grow trees *constrained* to have a nontrivial minimum weight per leaf. Of course, there are many other ways to handle this, including using the tree structure beyond the leaves; we just aim to conduct an exploratory evaluation here, as several of these areas remain ripe for future research.

# 4   Experimental Evaluation

We now turn to evaluating HEDGECLIPPER on publicly available datasets. Our implementation uses minibatch SGD to optimize (6), runs in Python on top of the popular open-source learning package `scikit-learn`, and runs out-of-core ($n$-independent memory), taking advantage of the scalability of our formulation. [3] The datasets are drawn from UCI/LibSVM as well as data mining sites like Kaggle, and no further preprocessing was done on the data. We refer to "Base RF" as the forest of *constrained* trees from which our implementation draws its specialists. We restrict the training data available to the algorithm, using mostly supervised datasets because these far outnumber medium/large-scale public semi-supervised datasets. Unused labeled examples are combined with the test examples (and the extra unlabeled set, if any is provided) to form the set of unlabeled data used by the algorithm. Further information and discussion on the protocol is in the appendix.

Class-imbalanced and noisy sets are included to demonstrate the aforementioned practical advantages of HEDGECLIPPER. Therefore, AUC is an appropriate measure of performance, and these results are in Table 2. Results are averaged over 10 runs, each drawing a different random subsample of labeled data. The best results according to a paired t-test are in bold.

We find that the use of unlabeled data is sufficient to achieve improvements over even traditionally overfitted RFs in many cases. Notably, in most cases there is a significant benefit given by unlabeled data in our formulation, as compared to the base RF used. The boosting-type methods also perform fairly well, as we discuss in the next section.

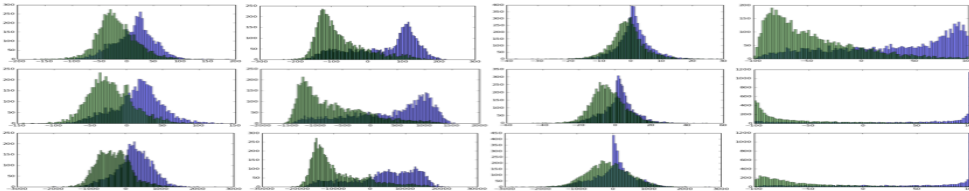

Figure 2: Class-conditional "awake ensemble prediction" ($\mathbf{x}^{\top}\sigma^{*}$) distributions, on SUSY. Rows (top to bottom): $\{1K, 10K, 100K\}$ labels. Columns (left to right): $\alpha = \{1.0, 0.3, 3.0\}$, and the base RF.

The awake ensemble prediction values $\mathbf{x}^{\top}\sigma$ on the unlabeled set are a natural way to visualize and explore the operation of the algorithm on the data, in an analogous way to the margin distribution in boosting [6]. One representative sample is in Fig. 2, on SUSY, a dataset with many (5M) examples, roughly evenly split between $\pm 1$. These plots demonstrate that our algorithm produces much more peaked class-conditional ensemble prediction distributions than random forests, suggesting margin-based learning applications. Changing $\alpha$ alters the aggressiveness of the clipping, inducing a more or less peaked distribution. The other datasets without dramatic label imbalance show very similar qualitative behavior in these respects, and these plots help choose $\alpha$ in practice (see appendix).

Toy datasets with extremely low dimension seem to exhibit little to no significant improvement from our method. We believe this is because the distinct feature splits found by the random forest are few in number, and it is the diversity in ensemble predictions that enables HEDGECLIPPER to clip (weighted majority vote) dramatically and achieve its performance gains.

On the other hand, given a large quantity of data, our algorithm is able to learn significant structure, the minimax structure appears appreciably close to reality, as evinced by the results on large datasets.

# 5   Related and Future Work

This paper's framework and algorithms are superficially reminiscent of boosting, another paradigm that uses voting behavior to aggregate an ensemble and has a game-theoretic intuition [1, 15]. There is some work on semi-supervised versions of boosting [16], but it departs from this principled structure and has little in common with our approach. Classical boosting algorithms like AdaBoost [17] make no attempt to use unlabeled data. It is an interesting open problem to incorporate boosting ideas into our formulation, particularly since the two boosting-type methods acquit themselves well

| Dataset | # training | HEDGECLIPPER | Random Forest | Base RF | AdaBoost Trees | MART [14] | Logistic Regression |
|---|---|---|---|---|---|---|---|
| kagg-prot | 10 | **0.567** | 0.509 | 0.500 | 0.520 | 0.497 | 0.510 |
| | 100 | **0.714** | 0.665 | 0.656 | 0.681 | 0.666 | 0.688 |
| ssl-text | 10 | **0.586** | 0.517 | 0.512 | 0.556 | 0.553 | 0.501 |
| | 100 | **0.765** | 0.551 | 0.542 | 0.596 | 0.569 | 0.552 |
| kagg-cred | 100 | **0.558** | 0.518 | 0.501 | 0.528 | 0.542 | 0.502 |
| | 1K | **0.602** | 0.534 | 0.510 | 0.585 | 0.565 | 0.505 |
| | 10K | **0.603** | 0.563 | 0.535 | 0.586 | 0.566 | 0.510 |
| a1a | 100 | **0.779** | 0.619 | 0.525 | 0.680 | 0.682 | 0.725 |
| | 1K | **0.808** | 0.714 | 0.655 | 0.734 | 0.722 | 0.768 |
| w1a | 100 | **0.543** | 0.510 | 0.505 | 0.502 | 0.513 | 0.509 |
| | 1K | 0.651 | 0.592 | 0.520 | **0.695** | **0.689** | 0.671 |
| covtype | 100 | **0.735** | 0.703 | 0.661 | 0.709 | **0.732** | 0.515 |
| | 1K | **0.764** | 0.761 | 0.715 | 0.730 | **0.761** | 0.524 |
| | 10K | 0.809 | **0.822** | 0.785 | 0.759 | 0.788 | 0.515 |
| ssl-secstr | 10 | 0.572 | **0.574** | 0.535 | **0.563** | 0.557 | 0.557 |
| | 100 | **0.656** | 0.645 | 0.610 | 0.643 | 0.637 | 0.629 |
| | 1K | 0.687 | 0.682 | 0.646 | **0.690** | **0.689** | 0.683 |
| SUSY | 1K | **0.776** | **0.769** | 0.764 | 0.747 | **0.771** | 0.720 |
| | 10K | 0.785 | **0.787** | **0.784** | **0.787** | **0.789** | 0.759 |
| | 100K | **0.799** | **0.797** | **0.797** | **0.797** | 0.796 | 0.779 |
| epsilon | 1K | 0.651 | 0.659 | 0.645 | 0.718 | 0.726 | **0.774** |
| webspam-uni | 1K | **0.936** | 0.928 | 0.920 | 0.923 | 0.928 | 0.840 |
| | 10K | 0.967 | **0.970** | 0.957 | 0.945 | 0.953 | 0.901 |

Table 2: Area under ROC curve for HEDGECLIPPER vs. supervised ensemble algorithms.

in our results, and can pack information parsimoniously into many fewer ensemble classifiers than random forests.

There is a long-recognized connection between transductive and semi-supervised learning, and our method bridges these two settings. Popular variants on supervised learning such as the transductive SVM [18] and graph-based or nearest-neighbor algorithms, which dominate the semi-supervised literature [8], have shown promise largely in data-poor regimes because they face major scalability challenges. Our focus on ensemble aggregation instead allows us to keep a computationally inexpensive linear formulation and avoid considering the underlying feature space of the data. Largely unsupervised ensemble methods have been explored especially in applications like crowdsourcing, in which the method of [19] gave rise to a plethora of Bayesian methods under various conditional independence generative assumptions on $\mathbf{F}$ [20]. Using tree structure to construct new features has been applied successfully, though without guarantees [21].

Learning with specialists has been studied in an adversarial online setting as in the work of Freund et al. [22]. Though that paper's setting and focus is different from ours, the optimal algorithms it derives also depend on each specialist's average error on the examples on which it is awake.

Finally, we re-emphasize the generality of our formulation, which leaves many interesting questions to be explored. The specialists we form are not restricted to being trees; there are other ways of dividing the data like clustering methods. Indeed, the ensemble can be heterogeneous and even incorporate other semi-supervised methods. Our method is complementary to myriad classification algorithms, and we hope to stimulate inquiry into the many research avenues this opens.

## Acknowledgements

The authors acknowledge support from the National Science Foundation under grant IIS-1162581.

## Footnotes

[1] We assume that (bounds on) the errors are, with high probability, true on the actual distribution. Such bounds can be derived using large deviation bounds or bootstrap-type methods.

[2] Since $\mathbf{b}$ is calculated from the training set and deviation bounds, we assume the problem feasible w.h.p.

[3]It is possible to make this footprint independent of $d$ as well by hashing features [13], not done here.

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
