[Reviews · NeurIPS 2015]

Submitted by Assigned_Reviewer_1

Given a set of classifiers and unlabeled data, the paper studied the problem how to minimize the worst-case expected classification error on the test data, and formulated the mathematics model (2). Compared to the original method [9], the authors also did several modifications, such as allowing abstain for each classifier on any instance x, and thus the unlabeled data matrix S could be sparse. The algorithm is described in Section 3.1, which builds a random forests, and uses Theorem 4 to estimate \delta^*.

1. As mentioned by the authors, the initial idea comes from [9], and I am worried about that whether this paper can still interest NIPS community after the publication of [9].

2. The method aims to optimize the worst case, but in practice, people are more interested in the performance in average case and the variance.

Others: a typo in line 117, change it to 'b\in (0,1]^p'.
Summary: This paper proposed a new algorithmic framework for improving the accuracy of classifier aggregation using unlabeled data. The initial idea comes from [9], and the main contributions of this paper are applying the idea in [9] to the context of random forests and showing its better performance in experiments.

Submitted by Assigned_Reviewer_2

The paper presents an algorithm to combine predictions from an ensemble of classifiers. It does this in a semi-supervised setting -- using the structure of the ensemble predictions on unlabeled data.

Experiments are reported on benchmark data sets with promising results.

The algorithm is based on a recent COLT paper [9]. The main contribution is to apply this theory to design an efficient and practical semi-supervised aggregation algorithm for random forests. More specifically, the main contribution is to extend the worst-case game-theoretic framework described in [9] to design an algorithm that narrows the gap between the worst-case scenario and reality thereby improving the performance of the algorithm on real/practical data sets. The framework described in [9] is extended using "specialists" that only predict on a subset of data and abstain from predicting on the rest.

This is an interesting, well-written paper. My main concern lies in the originality of the contribution. The "specialist aggregation game" proposed in this paper is a slight modification of the original game described in [9], where a different constraint set is used resulting in a slightly different optimization problem to estimate the weights needed to combine the ensemble of predictors.

It would be useful to report area under the precision-recall curve (along with AUC) for experiments with unbalanced data sets.
Summary: Interesting, well-written paper, but is heavily based on a recent COLT paper [9] and therefore scores low on originality.

Submitted by Assigned_Reviewer_3

This paper focusses on developing an efficient transductive ensemble learning algorithm that follows the theoretical work of [1]. The problem, that is characterized as a zero-sum game between a predictor and nature, is first reviewed. The authors then introduce a generalization to the scenario where the voters of the ensemble are "specialists" : voters that can abstain on a subset of the examples. An instanciation of the method to specialists learned using a Random Forest (named HedgeClipper) is devised from the theoretical framework, and empirically evaluated on real world datasets.

This paper is very well written and its original contributions are made very clear by the authors. The introduction and preliminaries are very complete, allowing the reader to understand the work of [1]. While tackling two out of four open problems stated in [1], the presented learning algorithm seems to be efficient, versatile, and fills in the gap between the theoretical framework of [1] and the practice.

The generalization to so-called "specialists" is interesting. The resulting learning algorithm based on Random Forests seems to be one of many ways to benefit from the introduced ideas, and open the way the new areas of investigation.

To conclude, this is a relevant generalization and instanciation of the novel theoretical work of [1]. I am unsure about the novelty aspects, but the work seem original and significant.

Typos and other minor comments : - Reference [5] appears before Reference [4], which should be correct by simply re-compiling. - Line 32: therefor -> therefore - Figure 1 is a bit blurry (on paper), and the {+1, -1} cells are not distinguishable when printed on paper. - Figure 2 is also a bit blurry. - Footnote numbers in the text have an extra space (before the number). - The source code is conveniently based on Scikit-Learn, and should be made publicly available after publication. - Line 514 of Supplementary Material has an extra space before the reference to Section 3.2.

[1] Akshay Balsubramani and Yoav Freund. Optimally combining classifiers using unlabeled data. In Conference on Learning Theory, 2015. To appear.

Update after rebuttal: I kept the same quality score.
Summary: This paper if of great quality and clarity, and seem to provide a significant generalization of the theoretical work of [1], together with an efficient learning algorithm based on this generalization.

Author Feedback
Author rebuttal: We thank each of the reviewers for their specific comments.

Reviewers 1-2 have reservations about originality of the theory with respect to the recent COLT paper [9]. However, our main contribution is to apply an algorithm inspired by that formulation to random forests, and to show that the results are better than random forests and other state-of-the-art ensemble methods for some large datasets. This is not possible with the original formulation of [9] (e.g. see line 200), necessitating our extension to specialists. As a nice side benefit, the specialist formulation is far more general than even [9] - for instance, it could be applied even to nearest-neighbor classifiers, or classifiers trained on only part of the data space, for which global error bounds make less sense.

The analytical simplicity of the specialist formulation is perhaps deceptive, because it is doing highly nontrivial learning. This is demonstrated by the experimental results (to reviewer 1, the worst-case assumptions lend structure to the problem without parameters, and are demonstrably not overly loose in the presence of enough specialists/constraints). Conceptually, we do not know of any algorithm that seamlessly accepts any pattern of specialist predictions without assumptions. This algorithm is learning the weights of an (almost) weighted majority vote from these patterns, which is very novel in our view.

The reviewers pointed out a few typos, which will be fixed; a couple of the figures will be redone. The references provided by R5 and R6 are very helpful (there are some flaws in [11], but we maintain that ensemble methods are dominant in achieving state-of-the-art performance e.g. as seen in competitions; we will add other references to support the point). The code will be made available after publication.